# Beam Training for Millimeter-Wave Communication Based on Tabu Table Enhanced Rosenbrock Algorithm

**Xiaoyu Li \*, Changyin Sun and Fan Jiang** 

Shaanxi Key Laboratory of Information Communication Network and Security, Xi'an University of Posts and Telecommunications, Xi'an 710121, China; sunchangyin@xupt.edu.cn (C.S.); fjiangwbc@gmail.com (F.J.)

**\*** Correspondence: xiaoyuli068@foxmail.com; Tel.: +86-1836-344-5558

**Abstract:** The codebook-based beamforming for millimeter-wave (mm Wave) communication systems is usually used to compensate the severe attenuation of the mm Wave region. The beam training process based on pre-specified beam codebooks is considered a global optimization problem in 2-D planes formed by the potential beam index. The Rosenbrock algorithm (RA) is adopted to implement optimum beam searching whereas the simulated annealing (SA) algorithm is used to solve the problem of falling into the local optimum, due to the unavailable gradient information of the objective function. However, the RA implements rounding to the integer which leads to the problem of repeated search and beam space discontinuity caused by beam index will impair the powerful local search ability. Thus, in this paper, an enhanced RA based on tabu search and combined with SA algorithm is proposed as an alternative solution for beam search success rate. The proposed algorithm reduces the search times by forbidding the repeat search with tabu table and design of neighbor region. Moreover, to prevent the search failure, the search candidate index is defined to keep the local search ability of the original algorithm and wrap around of beam index is applied to maintain continuity of the search direction. Experimental simulations show that the proposed technique can improve the search efficiency in terms of reduced steps and increase search success rate during the beam training procedure compared to existing techniques.

**Keywords:** millimeter-wave communications; beam training; tabu search

---

## 1. Introduction

In recent years, to meet the higher requirements on transmission rate and signal bandwidth, millimeter-wave (mm Wave) has been widely concerned because of its widely available bandwidth, strong anti-interference ability and short-wavelength [1]. Although mm Wave wireless communication has great application value, it faces many difficulties in practical applications due to its serious problems such as small coverage limited by severe path loss, intermittent transmission interrupted by human blockage and extra link budget due to absorption of the atmosphere [2]. To solve this problem, antenna array can be used to obtain remarkable processing gain, this technology can effectively compensate path loss and enhance the signal-noise ratio (SNR) of the receivers [3]. Concurrently, it can mitigate interference among users through directional transmission. In addition, multiple antennas can be expediently implemented on miniature portable devices, thanks to the millimeter wavelength. Usually, on the basis of the usage of the spatial domains, traditional multi-antenna technologies can be divided into two types, the spatial multiplexing and beamforming (BF). In order to easily implement and effectively improve the link budgets, it is widely believed that BF-based directional transmission is more appealing for mm Wave communications [4]. Therefore, BF has been adopted into the mm

---



Wave wireless communications related standards. On the other side, to establish and maintain the directional communication link between transmitter and receiver, beam search at both ends of the link is required, which always is much more time consuming and incurs a big burden on the system in terms of signaling overhead.

### 1.1. Related Work and Motivation

The existing beam search algorithms can be classified into three different categories: (1) exhaustive search; (2) hierarchical search [5]; and (3) compressive search [6]. In the exhaustive search scheme, both the transmitter and receiver sequentially scan all the beams to decide the best beam for transmission and reception. As the name suggests, it causes significant training overhead, as the searching times scale linearly with the number of the antenna. As an alternative, the hierarchical search scheme employs two search steps, at the first step, the beam is searched with a low-resolution codebook, based on the coarse direction fund at the first step, a finer search proceeds with a high-resolution codebook. Compared with the exhaustive search scheme, the hierarchical scheme saves much overhead at the cost of coverage loss, which due to the usage of a low-resolution codebook. In the compressive search scheme, the sparse scattering environments of the mm Wave system are considered, and compressive estimation is used to find the best beam. However, the compressive estimation requires the phase coherence across subsequent beacon, which is always difficult to implement [7].

On the contrary to the approaches assumed in the three schemes above, the beam training process is described as a limited space searching problem in the numerical optimization framework [8]. Modeled a combinatorial optimization of 2-D search space consisting of two pattern indexes from the specified beam codebook, a global direct search (GDS) algorithm is proposed to identify the best transmitter–receiver (Tx–Rx) beam-pair, using the minimum energy to maximize the receiving SNR [9]. In the literature [10], considering that the gradient information of the objective function (i.e., the receiving SNR) is practically unavailable, the Rosenbrock algorithm (RA) is used to carry out beam searching. By a two-step search implicitly approaching the function gradient, this algorithm typically exhibits promising search capabilities. However, when a non-smooth objective function containing many local optima is encountered, the Rosenbrock search may easily fall into local optima, which will cause the search failures. It is noteworthy that the receiving SNRs may exhibit many local optimum values, to overcome the shortcomings of the above classical RA, the literature [11] further proposed to combine RA with simulated annealing (SA) algorithm.

Although being effective in mitigating the problem of falling into local optimum, it is inevitable for RA that the intermediate search results are round to the integer to get the beam index, which will cause repeated search problems, moreover, considering the clustered structure of beam space immanent to mm Wave, both the correct mapping from beam space to search index, and processing of the index boundary will have a significant impact on the beam search performance. The tabu search (TS) is a kind of heuristic algorithm, which is used in combinatorial optimization problems to obtain the optimum solution. TS is required in many applications to solve optimization problems, including quadratic assignment problems (QAP) [12], detection problems [13], and GFBM systems [14], etc., for their efficiency. The TS algorithm relies on the concept of 'tabu', which prohibits the search of previous searched results to avoid falling into local maximum solution. The concept of 'tabu' motivates us to enhance the RA by TS algorithm.

### 1.2. Paper Contributions

In order to solve the above repeated search problems, we introduce TS to enhance the RA.

The main contributions of this paper can be summarized into three aspects: (1) The appropriate definitions of neighborhood structure, tabu table, aspiration criterion and stopping criterion involved in the TS-based RA are provided; (2) Based on the inherent clustered structure of beam space, design schemes of direction boundary and index boundary are developed. The former guarantees the continuity of the index at the direction boundary, and the latter guarantees the continuity of the search

direction at the index boundary; and (3) Inspired by the TS algorithm, an enhanced RA based on the tabu table is developed to solve the problem of repeated search, which is caused by rounding to the integer to get the beam index. The enhanced algorithm is proved by both theoretical analysis and experimental simulations. The results show that compared with the existing popular technologies, the proposed technology can considerably improve search success percentage and reduce the energy consumption during the beam search. The simulation results also show that this technology can achieve near optimal performance.

The paper is organized as follows. Section 2 will give a complete BF system model. Subsequently, the beam training based on RA is developed in Section 3. Then, the RA based on SA is introduced in Section 4. Section 5 is devoted to the enhanced RA based on tabu table. Experimental simulation and performance evaluation are given in Section 6. Finally, we conclude this article.

## 2. System Model

We consider the system model with BF as shown in Figure 1. Device 1 (DEV1) has $M_t$ transmit antennas, while device 2 (DEV2) employs $M_r$ receiver antennas. At the transmitter, the signal is multiplied by the transmission weight vector $w$, and is then emitted into wireless channels. Accordingly, the received signals at the receiver are multiplied by the received weight vector $c$, then combined together, and finally down-converted for baseband processing [15,16].

Considering the high-power consumption and the complicated realizations of RF electrical elements, beam training vectors with phase shift and amplitude adjustment seems to be impractical for mm Wave compatible devices. Accordingly, adopting a phased antenna array with fixed amplitude is a feasible alternative for rapid processing. The beam codebook can be denoted by $M \times N$ matrix **W**, which is specified by the number of elements $M$ and the required number of beams $N$. For a uniform linear array (ULA), the array response factor of the *nth* beam can be given by [17]

$$A_n(\theta) = \sum_{m=0}^{M-1} w_{m,\,n} e^{j2\pi m(d/\lambda)\cos\theta} \tag{1}$$

where $w_{m,\,n}$ corresponds the antenna weight factor of the *mth* antenna element of the *nth* codebook; $d$ is the antenna element spacing; and $\lambda$ is the wavelength, generally set $d = \lambda/2$. $\theta$ denotes the normal direction of the antenna array relative to *x_axis*.

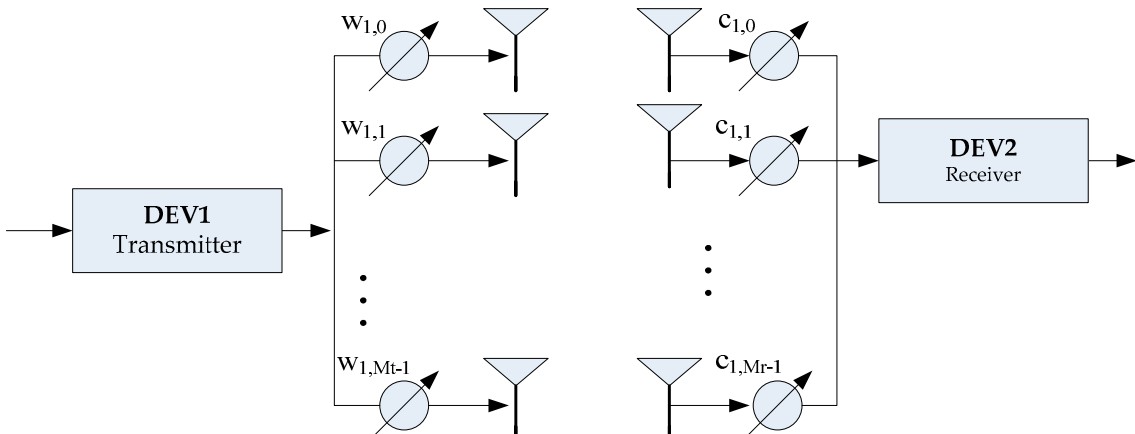

**Figure 1.** System model for beamforming (BF).

The channel impulse response (CIR) can be expressed as [18]

$$h(t,\ \theta,\ \phi) = \alpha_{0,\,0}\delta(t - \tau_{0,\,0})\delta(\theta - \theta_{0,\,0})\delta(\phi - \phi_{0,\,0}) + \sum_{l=1}^{L}\sum_{k=1}^{K_l}\alpha_{k,\,l}\delta\big(t - \tau_{k,\,l}\big)\delta\big(\theta - \theta_{k,\,l}\big)\delta\big(\phi - \phi_{k,\,l}\big) \quad (2)$$

where $L$ denotes the number of clusters; $K_l$ denotes the number of sub-paths of the *lth* cluster; $\alpha_{k,\,l}$ is the complex channel gain of the *kth* sub-path of the *lth* cluster; $\tau_{k,\,l}$ is the time delay; and $\theta_{k,\,l}$ and $\phi_{k,\,l}$ are the angle of arrival (AoA) and the angle of departure (AoD), respectively [19].

Accordingly, the SNR in the receiver can be written as:

$$SNR = \frac{1}{\sigma_n^2}\sum_{l=1}^{L}\sum_{k=1}^{K_l}\int_{\phi_{k,\,l}-\Delta\phi}^{\phi_{k,\,l}+\Delta\phi}\left|A_p\big(\theta_{k,\,l}\big)A_q(\phi)\right|^2\left|\alpha_{k,\,l}\right|^2 d\phi \quad (3)$$

where $A_p(\theta)$, $A_q(\phi)$ represent the beam array factors of the receiver and transmitter, respectively. $p$ and $q$ are the beam numbers selected for the receiver and transmitter, respectively (i.e., $n$ in the formula (1)). $\sigma_n^2$ is the noise power. $\Delta\phi$ denotes the effective spatial receiving range. Here, it is approximated that the array response factor $A_p\big(\theta_{k,\,l}\big)$ at the receiver is constant for each path. After BF, the transmitted energy will be concentrated. In the case of a direct path between devices, the propagation energy of the indirect path can be neglected for simplified analysis. The receiving SNR can be expressed as

$$SNR = \frac{1}{\sigma_n^2}\left(|\alpha_0|^2\int_{\phi_0-\Delta\phi}^{\phi_0+\Delta\phi}\left|A_p(\theta_0)A_q(\phi)\right|^2 d\phi\right) \quad (4)$$

In this paper, the receiving SNR is used as the objective function, with the beam index number of the transceiver as 2-D variables [20]. BF aims to obtain the optimum beam-pair $\big(p_{opt},\ q_{opt}\big)$ from the predefined codebook, to maximize the receiving SNR given by (4), i.e.,

$$\big(p_{opt},\ q_{opt}\big) = \max_{(p,\ q)} SNR(p,\ q), \quad (p,\ q) \in R^2 \quad (5)$$

It is useful to regard the beam training as a global optimization problem in a 2-D plane, which is formed by the transceiver and receiver beam-pair, i.e., $(p,\ q)$. The objective function is the receiving SNR, which is expressed as $snr(p,\ q)$.

## 3. Rosenbrock-Algorithm-Based Beam Training

### 3.1. Problem Description

Assume that the total number of beams at the transmitter and receiver are $N_t$ and $N_r$, respectively. For the most basic traversal search, the beam search will perform the $N_t \times N_r$ training sequence to determine the optimal beam-pair, which consumes a lot of energy.

From the system model analysis, the beam search problem can be transformed into the optimization problem of the 2-D discrete integer space, and the optimized objective function is the receiving SNR. However, the gradient information on the objective function is unknown in practice. The Rosenbrock search is very suitable for the optimization problem where explicit derivatives cannot be efficiently computed, but the objective function is not complicated to calculate. In this section, based on the RA, we develop an efficient beam training technology with evidently reduced overhead. To simplify the analysis, we employ the 1-D antenna array in the following detailed descriptions. With the beam index definition, the 1-D antenna array model can be easily extended to the 2-D antenna array case.

### 3.2. Rosenbrock Search

Instead of calculating the derivative, only simple evaluations of the objective function are required. The function gradient is then approached by a two-step search (i.e., probes and pattern move).

### 3.2.1. Probe Moving

In order to discover a new starting point and search direction, probe moving is performed along the $n$ orthogonal directions. In the analysis of this paper, $n = 2$ for the 1-D antenna array. Corresponding to two search directions, the initial move steps are denoted by $\xi_1$ and $\xi_2$, respectively. The initial solution during each probe round is given by $x^{(1)}$ and the solution along the *ith* (i.e., $i = 1, 2$) direction by $x^{(i+1)}$. Given the initial solution $s^{(1)} = \left(p^{(1)}, q^{(1)}\right)$, the initial directions $d^{(1)}$ and $d^{(2)}$, the magnification factor $\alpha \geq 1$, the shrinkage factor $\beta \in [-1, 0)$, the probe moving can be elaborated as follows.

Starting from $x^{(1)}$, we first probe along $d^{(1)}$. In addition, it is noteworthy that the beam indexes (i.e., $p$ and $q$) are integers. Accordingly, the output beam index used in the next probe moving should be rounded to the nearest integer. If we have $snr\left(x^{(1)} + round\left(\xi_1 d^{(1)}\right)\right) \geq snr\left(x^{(1)}\right)$, this probe operation is considered successful, and let

$$x^{(2)} = x^{(1)} + round\left(\xi_1 d^{(1)}\right) \tag{6}$$

where $round(x)$ denotes rounding the element $x$ to the nearest integer. The probe step is updated by $\xi_1 = \alpha \xi_1$, which leads to a larger movement in the next probe moves.

Otherwise, if $snr\left(x^{(1)} + round\left(\xi_1 d^{(1)}\right)\right) < snr\left(x^{(1)}\right)$, this probe operation is considered a failure, we set

$$x^{(2)} = x^{(1)} \tag{7}$$

In this case, the probe step is updated by $\xi_1 = \beta \xi_1$. Since $\beta$ is negative, a *back_off* search is employed in the next probe round. After probing along $d^{(1)}$, the similar operations will be implemented on $d^{(2)}$. Finally, the probed solution $x^{(n+1)}$ can be acquired and one round of probes is completed. The initial solution for the next round of probe iterations is set to $x^{(1)} = x^{(n+1)}$.

When all the direction moves fail, this probe movement iteration is terminated. Then, after the *kth* iteration, the new solution can be expressed as

$$s^{(k+1)} = x^{(n+1)} \tag{8}$$

### 3.2.2. Pattern Moving

From (8) and (6), we may have

$$s^{(k+1)} = s^{(k)} + \sum_{i=1}^{n} round\left(\lambda_i d^{(i)}\right) \tag{9}$$

where $\lambda_i$ represents the cumulative moving step along $d^{(i)}$.

It is observed that $p \overset{\Delta}{=} s^{(k+1)} - s^{(k)}$ may get close to the ascent direction, so during the next round of probes, the new constructed direction should take this ascent direction into consideration. The new constructed search directions can be formed from $s^{(k+1)} - s^{(k)}$ and further orthogonalized by utilizing the Gram-Schmidt orthogonalization procedure. Then, the algorithm alternates between the probe round and pattern round, until the relative change in the objective function values is below a pre-specified threshold $\eta$, i.e., $\|s^{(k+1)} - s^{(k)}\| \leq \eta$ [21].

Based on the above elaborations, the pseudo-code for the Rosenbrock search is provided in Algorithm 1.

### 3.2.3. Consideration of Limiting Factor

In practice, after many successful probes, the moving steps of the beam search may be unlimitedly amplified. If $snr\left(x^{(i+1)}\right) - snr\left(x^{(i)}\right) > \kappa \times 20 \log_{10} M$, the moving step will be limited to 1. Here, $\kappa \in (0, 1]$ is used to strengthen search efficiency and $M$ is the total number of antenna elements. However, the BF gain over different beam pairs exhibits a distinct clustered characteristic, so there are many local optimal values in the non-smooth objective function. Therefore, the Rosenbrock search may unavoidably fall into the local optimal values and terminate the search by eventually returning a non-optimal beam-pair [22].

---

**Algorithm 1.** Rosenbrock search.

---

**Input**: Initial solution $s^{(1)}$, $x^{(1)}$, and $s^{(1)} = x^{(1)}$; threshold $\eta$; initial counter $k = 1$.
**Output**: $\left(p_{opt}, q_{opt}\right)$.

1: Perform the probe moving process and output the probed solution $x^{(n+1)}$ (see Algorithm 2).
2: **if** $snr\left(x^{(n+1)}\right) \geq snr\left(x^{(1)}\right)$ **then**
3: $x^{(1)} = x^{(n+1)}$, and return to step 1.
4: **else**
5: Go to step 7.
6: **end if**
7: **if** $snr\left(x^{(n+1)}\right) \geq snr\left(s^{(k)}\right)$ **then**
8: $s^{(k+1)} = x^{(n+1)}$
9: **if** $\|s^{(k+1)} - s^{(k)}\| \leq \eta$ **then**
10: Output $s^{(k+1)}$.
11: **else**
12: $x^{(1)} = s^{(k+1)}$, perform the pattern moving process.
13: **end if**
14: **else**
15: $x^{(1)} = x^{(n+1)}$, and return to step 1.
16: **end if**
17: Output $\left(p_{opt}, q_{opt}\right)$.

---

## 4. Simulated-Annealing-Based Beam Training

### 4.1. Simulated Annealing

Simulated annealing (SA) is a generalized probability algorithm derived from the principle of solid annealing, which is used to find the optimal solution in a large search space. In traditional SA, the search starts at a higher temperature and accepts a solution that is worse than the current solution with a certain probability. In this way, the search can gradually jump out of the local optimum. The current state of the thermodynamic system is similar to the current solution; the energy equation of the thermodynamic system is similar to the objective function (i.e., the receiving SNR). Therefore, the basic elements of the SA algorithm in our beam training can be described in details as follows.

Firstly, the objective function that represents the state energy should be defined, i.e., $E(i) = snr\left(x^{(i)}\right)$. In this paper, the probability of switching from the current solution $x^{(i)}$ to a new candidate solution $x^{(i+1)}$ is specified by an acceptance probability $P(E(i), E(i+1), T)$, which depends on two energy amplitudes $E(i) = snr\left(x^{(i)}\right)$ and $E(i+1) = snr\left(x^{(i+1)}\right)$ and a time-varying parameter $T$. Given the

current temperature $T$, the probability of a decrease in energy amplitude (i.e., $\Delta E = E(i) - E(i+1) > 0$) can be defined by

$$P(\Delta E,\, T) = \frac{\exp(-E(i)/kT)}{\exp(-E(i)/kT) + \exp(-E(i+1)/kT)} \approx \exp(-\Delta E/kT) \qquad (10)$$

where $k$ denotes the Boltzmann's constant. If $\Delta E > 0$, the state transition from $x^{(i)}$ to $x^{(i+1)}$ is accepted. Otherwise, if $\Delta E < 0$, the new solution $x^{(i+1)}$ may be allowed to use the probability returned by (10). The state transition happens if the following condition is fulfilled:

$$P(\Delta E,\, T) \geq \gamma \qquad (11)$$

where $\gamma$ denotes a random number between 0 and 1, i.e., $\gamma \sim U(0,\, 1]$.

In addition to the high initial temperature $T_0$, a gradual reduction of temperature is also required as the search progresses. So, we have:

$$T(k) = T_0 \times \kappa^k \qquad (12)$$

Here, $k$ is the updating iterator. The decay factor $\kappa$ satisfies $0 < \kappa 1$.

### 4.2. Simulated Annealing-Based Rosenbrock Search

In order to solve the problem of falling into the local optimum, the SA algorithm is added to the Rosenbrock search, which applies the probabilistic acceptance criteria to the probe movement. Specifically, two temperature parameters $T_l$ and $T_g$ are adopted in this method, i.e., the local temperature and the global temperature, which correspond to the probe moving and the pattern moving, respectively. The temperature parameter $T_g$ is updated after the new search directions have been constructed, while $T_l$ is updated after each probe moving stage.

Despite the fact that, the combination of the SA algorithm with the RA makes it possible to avoid falling into the local optimum because of the probabilistic property of the SA algorithm. It should be emphasized that certain problems still remain unsolved due to the intrinsic features of the beams search. For example, practical beam indexes are both integers, while the output beam indexes for the next probe moving should be rounded to the nearest integer. Simulation analysis found that rounding to the nearest integer makes the algorithm suffer the repeat search problems. The repeated search problem not only consumes a lot of search steps but also cause unavoidable search failure.

## 5. Tabu-Search-Based Enhanced Rosenbrock Algorithm

Tabu Search (TS) algorithm was proposed by Glover. TS is characterized by the use of tabu tables to block access to recently searched solutions [23]. This feature solves the repeated search problem found in the previous section. Given that the mm Wave channel is sparse and clustered, the beam gain exhibits clustered peaks at distinct local positions in beam index space, which represents the directions of the corresponding rays. To make full use of the local search ability of RA when combined with TS algorithm, mapping from the beam index to search index for search need to be defined.

### 5.1. Beam Index Definition

In this paper, the beam search is performed in a 2-D discrete integer search index space formed by the model of the Tx–Rx beam index. Both uniform linear arrays (ULA) and uniform planar arrays (UPA) will be considered in the following analysis. In the case of UPA, both the horizontal beam direction and the vertical beam direction should be considered for beam transmission and reception. In order to implement beam search in the 2-D beam index plane, we need to define mapping from the beam index to search index.

Considering the inherent sparse and clustered structure of the beam space, and assuming that the beam is transmitted along the $h_t th$ horizontal direction and the $v_t th$ vertical direction, the search index is denoted by the mapping function of $I_t = index(v_t, h_t)$ as follows:

$$I_t = index(v_t, h_t) = \begin{cases} (v_t - 1) \cdot H_t + (-1)^{(v_t-1)} \cdot h_t, & \text{if } v_t \text{ is odd} \\ v_t \cdot H_t + (-1)^{(v_t-1)} \cdot (h_t - 1), & \text{if } v_t \text{ is even} \end{cases} \tag{13}$$

The receiver search index $I_r$ is defined by this method similarly.

The definition of the search index is shown as an example in Table 1, where we set the number of the horizontal beam directions $H_t = H_r = 32$, and the number of the vertical beam directions $V_t = V_r = 4$.

**Table 1.** Description of the beam index.

| | 1 | ... | $h_t$ | ... | 32 |
|---|---|---|---|---|---|
| $v_t = 1$ | 1 | ... | $I_t = (v_t - 1) \cdot H_t + (-1)^{(v_t-1)} \cdot h_t$ | ... | 32 |
| $v_t = 2$ | 64 | ... | $I_t = v_t \cdot H_t + (-1)^{(v_t-1)} \cdot (h_t - 1)$ | ... | 33 |
| $v_t = 3$ | 65 | ... | $I_t = (v_t - 1) \cdot H_t + (-1)^{(v_t-1)} \cdot h_t$ | ... | 96 |
| $v_t = 4$ | 128 | ... | $I_t = v_t \cdot H_t + (-1)^{(v_t-1)} \cdot (h_t - 1)$ | ... | 97 |

*5.2. Tabu Search*

The basic idea of the proposed RA enhanced by the tabu table is described in this section. Specifically, the most recent searched solutions of RA is recorded in the tabu table, during the probe moving round, if a probed solution is in the tabu table after rounding to the integer operation, the tabu search is applied to obtain the solution, which optimally is near the solution before rounding operation, but different with the solution after the rounding operation. The proposed algorithm proceeds as follows. Firstly, the probed solution $x^{(i+1)}$ is obtained by exploring the *ith* direction of the probe moving just like in the classical RA. If $x^{(i+1)}$ is in the tabu table, the best solution among the neighborhood of $x^{(i+1)}$ is selected as the starting point for the next move, and using the special criteria to reward some excellent states. In the next section, four important aspects of the RA enhanced by tabu table, including neighborhood definition, tabu table, aspiration criterion, and stopping criterion, will be elaborated in detail.

1.  Neighborhood definition

Neighborhood structure and size are vital parameters for the TS algorithm to improve performance. A larger neighborhood will reduce the time TS stays in a local optimal state and accelerate its movement towards the global optimum. However, if the neighborhood is too large, the time complexity will make the TS algorithm infeasible. In this paper, taking the current solution $x^{(i)}$ as the center, and the open interval with $R$ as the radius is called the neighborhood of $x^{(i)}$, which is denoted as $v_R(x^{(i)})$. The neighborhood radius of 1 as an example is discussed in this paper. In TS, defined neighborhoods are not all searched, but can be a part of them. We can increase the scope of exploration by increasing the radius of the neighborhood, but the search complexity increases. To this end, some methods are adopted to select some neighborhood explorations, such as random selection, or steepest descending direction selection. For UPA with $R = 1$, the horizontal direction and vertical direction corresponding to $v_R(x^{(i)})$ satisfies: (1) It has only one column that is different from the corresponding column in $x^{(i)}$; (2) The direction difference between the two corresponding columns equals one.

Let $x^{(i)}$ denotes the current solution of the proposed TS-based beam search, and $v_1(x^{(i)}) = \{U_1^{(i)}, U_2^{(i)}, \ldots, U_{|v_1|}^{(i)}\}$ presents the neighborhood of $x^{(i)}$, where $|v_1|$ is the cardinality of $v_1$. Based on the neighborhood definition above, it is obvious that $|v_1| = 2 \cdot 2^n = 2^{n+1}$. We then define that the *uth* neighbor in $v_1(x^{(i)})$ is different from $x^{(i)}$ in the $\lceil u/2^n \rceil th$ column [24]. According to the index definition

above, it is obvious that $x^{(i)} = [index(v_t, h_t), index(v_r, h_r)]$, which shows that in the current solution, the signal is transmitted along the $h_t th$ horizontal direction and the $v_t th$ vertical direction, and is received along the $h_r th$ horizontal direction and the $v_r th$ vertical direction. Both $v_t$ and $v_r$ are odd numbers. Take this case as an example, the neighborhood of $x^{(i)}$ is:

$$
\begin{aligned}
U_1^{(i)} &= [index(v_{t-1}, h_t), index(v_r, h_r)], \ U_2^{(i)} = [index(v_{t+1}, h_t), index(v_r, h_r)] \\
U_3^{(i)} &= [index(v_t, h_{t-1}), index(v_r, h_r)], \ U_4^{(i)} = [index(v_t, h_{t+1}), index(v_r, h_r)] \\
U_5^{(i)} &= [index(v_t, h_t), index(v_{r-1}, h_r)], \ U_6^{(i)} = [index(v_t, h_t), index(v_{r+1}, h_r)] \\
U_7^{(i)} &= [index(v_t, h_t), index(v_r, h_{r-1})], \ U_8^{(i)} = [index(v_t, h_t), index(v_r, h_{r+1})]
\end{aligned}
\tag{14}
$$

2. Tabu table

The tabu table $T(x)$ is a container for storing tabu solutions, so it is not feasible to search for this solution placed in the tabu table until the solution is removed. During the tabu search process, the first step is to select the best solution among the neighborhood $v_R(x^{(i)})$ as the starting point for the next iteration, then, in the second step, the algorithm determines whether the new solution is in $T(x)$. The solutions in the tabu table are updated each time when the tabu search iterate once, the solutions in the tabu table are updated in a FIFO style, the nearest solution is placed at the blank top of the $T(x)$, and the earliest solution is released from the $T(x)$.

Let us denote $l_{avg}$ and $|T(x)|$ as the average number of iterations before the same solution vector reappears, and the length of the tabu table. The size of $l_{avg}$ and $|T(x)|$ will affect the performance of the algorithm. If the size is too large, the complexity will be increased. If it is too small, the success rate will be reduced. $l_{avg}$ and $|T(x)|$ will be set to constants through simulation experiments. Let us define the tabu period $S$, a non-negative integer parameter, as follows: (1) The new solution vector has not been previously searched. In this case, the value of variable $S$ is not updated. (2) The new solution vector has been searched previously. We update the value of variable by comparing with $S$. Also, the value of $S$ is updated as follows:

$$
S = \begin{cases} S+1, & S \le l_{avg} \\ S-1, & S > l_{avg} \end{cases}
\tag{15}
$$

When the tabu period $S$ of the tabu object is non-zero, the corresponding tabu object is prohibited [14].

3. Aspiration criterion

The aspiration criterion needs to define to be defined based on an aspiration function $A(x)$. In this paper, the object value of the best solution has been obtained before the current iteration is selected as the aspiration function. When aspiration criterion is meet: $snr(x^{(i)}) > A(x)$, even if $x^{(i)}$ is in the $T(x)$, we still select it as the starting point for the next iteration.

4. Stopping criterion

We define *iter* as a parameter to indicate how long (in terms of the number of iterations) the global optimal solution has not been updated. Based on the parameter *iter*, the algorithm iteration will be stopped when the number of iterations reaches the pre-defined maximum number of iterations *max_iter*, i.e., *iter* ≥ *max_iter* [25].

*5.3. Boundary Problems*

5.3.1. Neighborhood Boundary Description

With the neighborhood definition in (14), the neighborhood may lay beyond the index interval near the boundary, i.e., $v_{t-1} < 1$ or $v_{t+1} > V_t$ and $h_{t-1} < 1$ or $h_{t+1} > H_t$. To remedy the boundary

problem of the neighborhood, we suppose the vertical direction in the neighborhood as $v^*$ and the horizontal direction in the neighborhood as $h^*$, when $v^* < 1$ or $v^* > V$, we have

$$v^* = \text{mod}(v^*, V) \tag{16}$$

when $h^* < 1$ or $h^* > H$, we have

$$h^* = \text{mod}(h^*, H) \tag{17}$$

### 5.3.2. Index Boundary Description

Since the beam search is performed in the 2-D discrete integer plane, we should ensure that the beam index is always within the index range during the probe moving. As shown in Figure 2, the beam index exhibits intrinsic structure at the border of the search region. Considering that the RA is essentially a local search, in order to solve the boundary problem of the search area, we surround the beam indexes when the index range is exceeded. Taking the beam index $I$ as an example, the beam index beyond the range (i.e., $I < 1$ or $I > N$) is given by

$$I = \begin{cases} \text{mod}(I, N), & if\ \text{mod}(I, N) \neq 0 \\ N, & otherwise \end{cases} \tag{18}$$

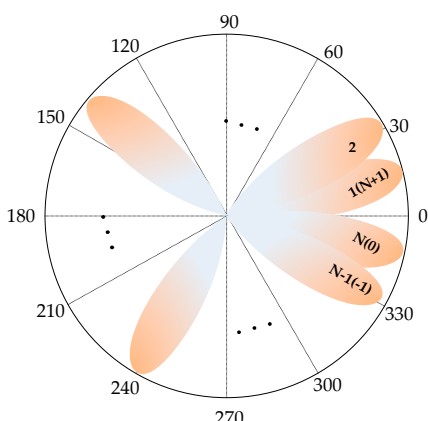

**Figure 2.** Beam diagram.

Based on the earlier discussions on the four important aspects of TS, we apply the TS algorithm to the probe moving of the RA in Algorithm 1. The flow of the probe moving of the proposed beam training algorithm is summarized in Algorithm 2. Notice that: (1) Algorithm 2 is corresponding to subprogram of Step1 in Algorithm 1; (2) the SA algorithm is combined with Rosenbrock, so temperature parameter $T_g$ is updated in the pattern moving in Algorithm 1; (3) the complete algorithm flow of the suggested beam search scheme which is based on the newly designed numerical search has been provided in Algorithms 1 and 2.

---

**Algorithm 2.** The probe moving of the proposed algorithm (Subprogram of Step1 in Algorithm1).

---

**Input**: Initial solution $x^{(1)}$; aspiration function $A(x)$; initial move steps $\xi_1$, $\xi_2$; initial search directions $d^{(1)}$, $d^{(2)}$; tabu table $T(x)$; initial iteration counter *iter* and the maximum iterations *max_iter*.
**Output**: The probed solution $x^{(n+1)}$.

1:     **for** $i = 1 - 2$ do
2:     **if** the beam index beyond the range **then**
3:     Calculate the beam index according to (18).
4:     **end if**
5:     **if** $snr\big(x^{(i)} + round\big(\xi_i d^{(i)}\big)\big) > snr\big(x^{(i)}\big)$ **then**
6:     $x^{(i+1)} = x^{(i)} + round\big(\xi_i d^{(i)}\big)$
7:     **if** $x^{(i+1)} > A(x)$ **then**
8:     $A(x) = x^{(i+1)}$
9:     **end if**
10:    Update $T(x)$.
11:    **else**
12:    **while** $iter < $ max_iter **do**
13:    Update $x^{(i+1)}$ and $T_l$ according to SA.
14:    Update $T(x)$ and $A(x)$ according to TS.
15:    **end while**
16:    **end if**
17:    **end for**
18:    Output $x^{(n+1)}$.

---

### 5.3.3. Computational Complexity and Limitation

The computational complexity and limitation of the proposed algorithm is investigated in this section. Compared with the classical RA, extra tabu search is required only when rounding to the integer brings the result to a solution searched previously. Moreover, some storage resources are also required to memory the tabu table. In the lower bound case, the increased search times are $max\_iter \times 1$, and in the upper bound case, the increased search times are $max\_iter \times R$. The main factor which has much effect on the complexity is the calculation of the object, i.e., the $snr(p, q)$. The calculation includes $M$ times multiplication of the signal by the transmission weight vector $w$ at the transmitter, and $N$ times multiplication of the signal by the receive weight vector $c$ at the receiver. In total, the computation complexity is $max\_iter \times (M + N)$ in terms of complex multiplication in the lower bound case, and $max\_iter \times (M + N) \times R$ in the upper bound case. Clearly, the computation complexity scale linearly with the number of antenna elements at both the transmitter and receiver. However, the cost incurred by the increased computation complexity can make up for the cost saved by the decreased computation complexity due to the forbidden repeated search.

Although the proposed methods are promising for BF in mm Wave, considering that it is essentially a kind of direct search optimization algorithm, the proposed algorithm is more beneficial when the objective function exhibits some kind of smoothness. In other words, the optimum parameters of the method may be influenced by the mm Wave environment, and should be carefully designed, we leave this as our future research work.

## 6. Experimental and Result Evaluations

In this section, we will evaluate the performance of our proposed algorithm by simulations. The mm Wave channel model by [26] is used. For analysis convenience, the 1-D uniformly spaced antenna array is used during the simulation experiments. The antenna array element number $M$ is assumed to be 64 (i.e., $M_t = M_r = M$).

### 6.1. Beam Training Performance Evaluation

Figure 3 shows the received SNR of three algorithms based on 100 independent experiments. The three algorithms are classical RA, RA based on SA ($RA + SA$), and enhanced RA based on TS ($RA + SA + TS$), respectively. The performance of the proposed algorithm is evaluated by the achievable values of receiving SNR. The upper bound values are the optimum SNR by exhausted search. According to statistical calculations of the simulation results, the search success probability of the classical RA is 36%; the RA based on SA is 60%, and the enhanced RA based on TS is 92%. As shown in Figure 3, the searched values are the most consistent with the target when using the enhanced RA during beam training. Although the tabu table introduces some complexities due to the storage of solutions, the increase in the complexity of the enhanced algorithm is not great compared with the existing algorithms, which is of great significance for practical applications. Therefore, the enhanced algorithm proposed in this paper is more conducive to the actual optimization problem.

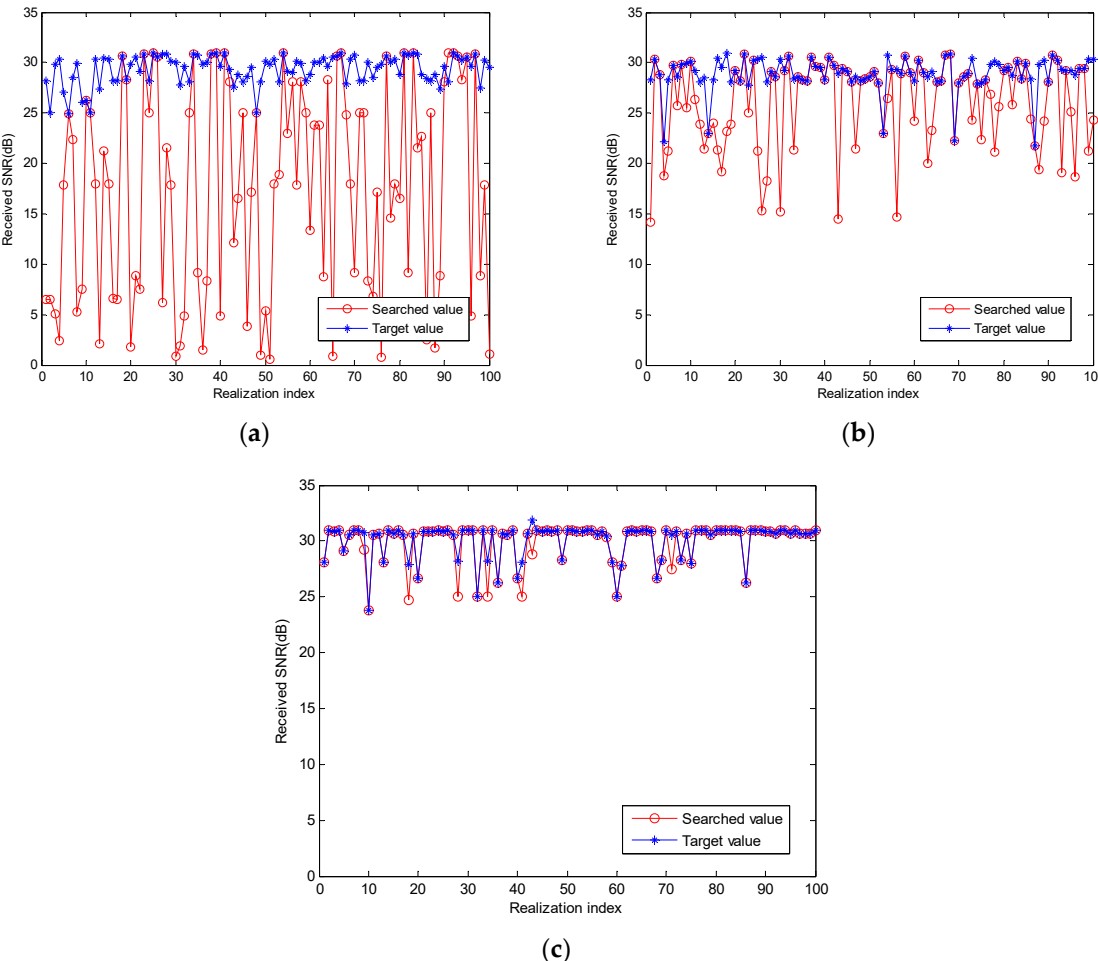

**Figure 3.** Performance of the Rosenbrock search in mm Wave beam training. (**a**) Classical Rosenbrock algorithm; (**b**) $RA + SA$; (**c**) $RA + SA + TS$ algorithm.

### 6.2. Boundary Processing

In order to ensure that the beam search is fast and efficient, the boundary problem described in detail in Section 5 is solved by the wrapping around method, including the neighborhood boundary and the probe boundary. The wrap around at the neighborhood boundary is applied to ensure the continuity of the beam index, and the wrap around at the beam index is applied to ensure the continuity of the search direction. In Figure 4, we plot the results of 100 independent implementations in both classical technique (i.e., without wrapping around) and wrapping around. From the performance

analysis method above, it can be seen that applying the wrap around method can greatly improve the performance of the algorithm, so it is necessary to apply the wrap around method to solve the problem of the boundary.

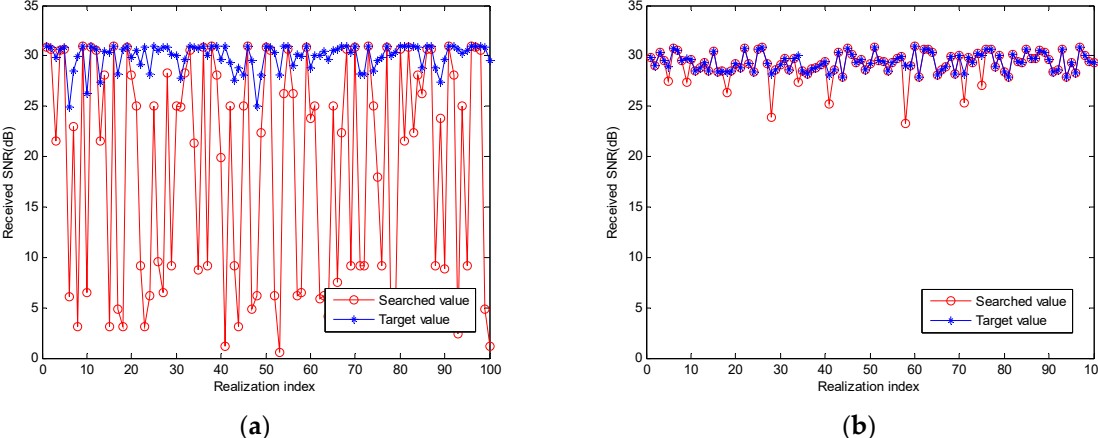

**Figure 4.** Performance of the boundary definition method. (**a**) Classical technique; (**b**) Wrapping around of the boundary.

### 6.3. Neighborhood Structure

In TS, neighborhood structure and size are key factors for a local search algorithm to get significant performance. The larger the neighborhood is, the higher the probability of successful search, the more energy will be consumed. We compare the search performance of different neighborhood sizes (i.e., radius of the neighborhood $R$) with the method of neighborhood structure as defined in Section 5. As shown in Figure 5, based on the 100 independent realizations, we have plotted the cumulative distribution function (CDF) diagram of searched values at different neighborhood radius $R$ in order to make the performance comparison more clearly. We get the best search performance when $R = 2$. Moreover, the analysis shows that the complexity of $R = 2$ is lower than $R = 3$. So, in this paper, the neighborhood radius is set to 2.

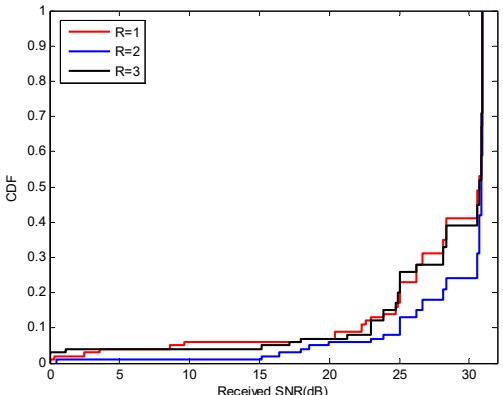

**Figure 5.** The cumulative distribution function (CDF) diagram of search results at different neighborhood radius.

### 6.4. Complexity Analysis

In Figure 6, we present the experimental results of the complexity of 100 independent implementations of classical RA and enhanced RA. The complexity is in terms of search steps. The experimental results show that the proposed scheme brings certain complexity increase compared with the traditional scheme, because the proposed scheme evaluates the objective function at the solution set in the neighborhood.

However, it is emphasized that, such increase in complexity can make up for the increased complexity of the repeated search brought by the traditional scheme. The proposed algorithm greatly increases the probability of successful searches at the cost of moderate complexity increase. With the further increase in the number of elements, the proposed algorithm can still be implemented efficiently, which is of great significance for practical applications.

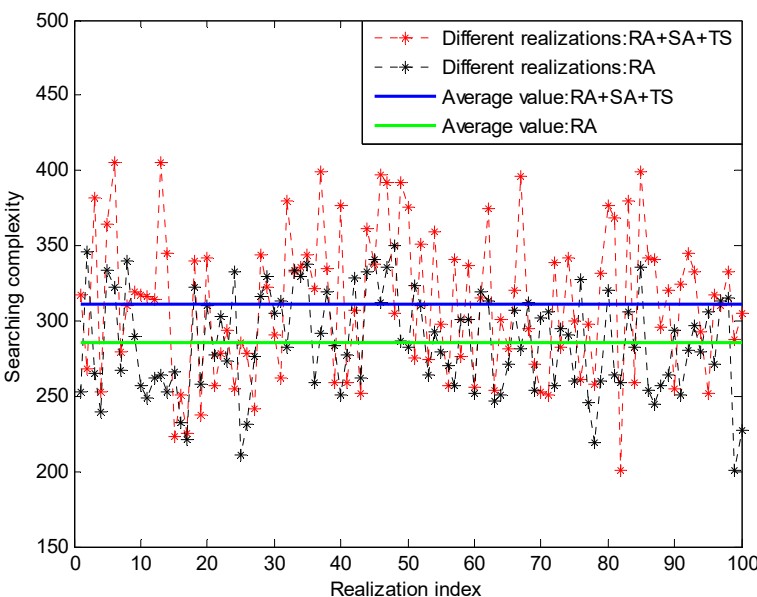

**Figure 6.** Search performance of the Rosenbrock algorithm (RA) and the new numerical algorithm (RA + SA + TS).

## 7. Conclusions

In this paper, we formulate the beam training process as a finite-space search problem. To solve the repeated search problem in the previous research caused by round to the near integer, a search algorithm based on tabu table is proposed in order to overcome the shortcomings of existing search algorithms. In addition, we also discuss the boundary problem of the beam index and the neighborhood structure of the tabu algorithm. As the proposed algorithm is a combination of the RA and SA, and further enhanced by TS, so it can implicitly approach the function gradient, jump out of the local optimum, and avoid the performance of repeated searches. Simulations prove that it can improve the probability of successful search. In the future, we will address the impact of the initial value on the search performance, which is important to beam training.

**Author Contributions:** Conceptualization, X.L.; Data curation, X.L.; Formal analysis, X.L.; Resources, C.S.; Supervision, C.S. and F.J.; Visualization, X.L.

**Funding:** This research was funded by the National Natural Science Foundation of China (No. 61801382, 61871321), National Science and Technology Major Project of the Ministry of Science and Technology of China (project number: ZX201703001012-005). Key Project of Natural Science Foundation of Shaanxi Province (project number: 2019JZ-06). Key Industrial Chain Project of Shaanxi Province (project number: 2019ZDLGY07-06).

**Acknowledgments:** The authors thank the fund of National Natural Science Foundation of China (No.61801382, 61871321) and the fund of National Science and Technology Major Project of the Ministry of Science and Technology of China (project number: ZX201703001012-005) for covering the costs to publish in open access and the costs when writing this study. Besides, the authors thank the anonymous reviewers for their insightful comments that helped improve the quality of this study.

**Conflicts of Interest:** The authors declare no conflict of interest.

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
