# Peer review of "Beam Training for Millimeter-Wave Communication Based on Tabu Table Enhanced Rosenbrock Algorithm"

_futureinternet, doi:10.3390/fi11100214_

Round 1
Reviewer 1 Report
The authors claim to propose a new variation of the rosenbrock algorithm to enhance the beam search success rate by solving the limitations of the original rosenbrock algorithm. The proposed algorithm is based on a tabu table approach. They also claim that their results can improve the search efficiency in terms of reduced steps and increase search success rate during the beam training procedure. The algorithm’s performance was compared against current solutions.
Even though the manuscript’s contribution is not a breakthrough in the field, the manuscript provides enough convincing evidence for its conclusions and might be of general interest for the community. Therefore, considering the facts previously mentioned, I recommend this manuscript for publication if the authors address the issues described below (see items 1-9 please) before final publication. Extensive editing of English language is also strongly recommended to increase the manuscript’s quality. Hence, please, proofread the manuscript carefully. I kindly rewrote the abstract and strongly recommend the authors to use it.
Items 1 to 12
Suggested abstract:
The codebook-based beamforming for millimeter-wave communication systems is usually used to compensate the severe attenuation of the mm Wave region. The beam training process based on pre-specified beam codebooks is considered a global optimization problem in 2-D planes formed by the potential beam index. The rosenbrock algorithm (RA) is adopted to implement optimum beam searching whereas the simulated annealing (SA) algorithm is used to solve the problem of falling into the local optimum, due to the unavailable gradient information of the objective function. However, the RA implements rounding to the integer which leads to the problem of repeated search and beam space discontinuity caused by beam index will impair the powerful local search ability.
Thus, in this paper, an enhanced RA based on tabu search and combined with SA algorithm is proposed as an alternative solution for beam search success rate. The proposed algorithm reduces the search times by forbidding the repeat search with tabu table and design of neighbor region. Moreover, to prevent the search failure, the search candidate index is defined to keep the local search ability of the original algorithm and wrap around of beam index is applied to maintain continuity of the search direction. Experimental simulations show that the proposed technique can improve the search efficiency in terms of reduced steps and increase search success rate during the beam training procedure compared to existing techniques.
“Although mm Wave wireless communication has great application value, it faces many difficulties in practical applications due to its serious path loss and other problems.”
Which other problems? How much is the path losses? Please include a reference to this affirmation.
How much computational efforts the proposed algorithm require to converge to the global optimum?
How does the proposed algorithm behaviour in terms of convergence and computational efficiency when the number of elements M and the required number of beams N become very large? How about when the total number of beams at the transmitter and receiver become very large? How about when the table search becomes extremely large?
"Simulations prove that it can improve the probability of successful search"
How much % can it improve?
Include the unit of the X-axis of Figure 5. The tabu table based-search approach adds some complexity to the proposed algorithm. Can you elaborate on it? How much is the complexity increased? How about the storage of solutions? How does the proposed algorithm deal with larger neighbourhood? What are the restrictions/limitations/time? Can the authors elaborate on it? The resolution/quality of the plotted graphics is quite low. Please try to use vector graphic images. Make sure the references follow the standard template of the journal
Author Response
Response to Reviewer 1 Comments
Point 1: Extensive editing of English language is also strongly recommended to increase the manuscript’s quality. 

Response 1: We have edited the language and style of the article extensively. (in red)
Point 2: I kindly rewrote the abstract and strongly recommend the authors to use it.
Response 2: (In Abstract section) The reviewer 1 has helped us to rewrite the abstract, we thought it was necessary. We adopted the reviewer 1's suggestion abstract. Thanks to reviewer 1! (in purple)
Point 3: The introduction needs to provide sufficient background and include all relevant references.
Response 3: (In Introduction section) We've rewritten the introduction. Please refer to the “Introduction” section. We provided references [5-7], [12-14]. (in purple)
Point 4: “Although mm Wave wireless communication has great application value, it faces many difficulties in practical applications due to its serious path loss and other problems.” Which other problems? How much is the path losses? Please include a reference to this affirmation.
Response 4: (In Introduction section) We've rewritten the first paragraph. We removed the following statement “Although mm Wave wireless communication has great application value, it faces many difficulties in practical applications due to its serious path loss and other problems.” We added the following statement “Although mm Wave wireless communication has great application value, it faces many difficulties in practical applications due to its serious problems such as small coverage limited by severe path loss, intermittent transmission interrupted by human blockage and extra link budget due to absorption of the atmosphere [2].“. We provided reference [2]. (in purple)
Point 5: How much computational efforts the proposed algorithm require to converge to the global optimum?
Response 5: (In Tabu-Search-Based Enhanced Rosenbrock Algorithm section) We added “Computational complexity and limitation” section to explain the Point 7.
(In Experimental and Result Evaluations section) We have further added “Complexity Analysis” section to compare the searching complexity of different algorithms. (in purple)
Point 6: How does the proposed algorithm behaviour in terms of convergence and computational efficiency when the number of elements M and the required number of beams N become very large?
Response 6: (In Tabu-Search-Based Enhanced Rosenbrock Algorithm section) We added the following statement “In total, the computation complexity is in terms of complex multiplication in the lower bound case, and in the upper bound case. Clearly, the computation complexity scale linearly with the number of antenna elements at both the transmitter and receiver.“. Please refer to the “Computational complexity and limitation” section. (in purple)
Point 7: How about when the total number of beams at the transmitter and receiver become very large?
Response 7: The total number of beams at the transmitter and receiver are the required number of beams N. We have already answered this question in Point 6.
Point 8: How about when the table search becomes extremely large?
Response 8: (In Tabu-Search-Based Enhanced Rosenbrock Algorithm section) We added the following statement to explain the Point 8: “Let us denote and as the average number of iterations before the same solution vector reappears, and the length of the tabu table. The size of and will affect the performance of the algorithm. If the size is too large, the complexity will be increased. If it is too small, the success rate will be reduced. and will be set to constants through simulation experiments.” We’ve defined the length of the tabu table as a constant, so the size of the tabu search is constant. Please refer to the “Tabu Table” section. (in purple)
Point 9: "Simulations prove that it can improve the probability of successful search" How much % can it improve?
Response 9: (In Experimental and Result Evaluations section) We've written them in the second paragraph: “The performance of the proposed algorithm is evaluated by the achievable values of receiving SNR. The upper bound values are the optimum SNR by exhausted search. According to statistical calculations of the simulation results, the search success probability of the classical RA is 36%; the RA based on SA is 60%, and the enhanced RA based on TS is 92%. As shown in Figure 3, the searched values are the most consistent with the target when using the enhanced RA during beam training.” Therefore, the success rate of the enhanced RA based on TS we studied is 56% higher than the classical RA, and 32% higher than the RA based on SA. Please refer to the second paragraph in the “Experimental and Result Evaluations” section. (in purple)
Point 10: Include the unit of the X-axis of Figure 5.
Response 10: We’ve successfully modified Figure 5. Please refer to the “Experimental and Result Evaluations” section. (in purple)
Point 11: The tabu table based-search approach adds some complexity to the proposed algorithm. Can you elaborate on it? How much is the complexity increased?
Response 11: (In Tabu-Search-Based Enhanced Rosenbrock Algorithm section) We added the following statement “The computational complexity and limitation of the proposed algorithm is investigated in this section. Compared with the classical RA, extra tabu search is required only when rounding to the integer bring the result to a solution searched previously. Moreover, some storage resources are also required to memory the tabu table. In the lower bound case, the increased search times are , and in the upper bound case, the increased search times are . The main factor which has much effect on the complexity is the calculation of the object, i.e. the . The calculation includes times multiplication of the signal by the transmission weight vector at the transmitter, and times multiplication of the signal by the receive weight vector at the receiver. In total, the computation complexity is in terms of complex multiplication in the lower bound case, and in the upper bound case. Clearly, the computation complexity scale linearly with the number of antenna elements at both the transmitter and receiver. However, the cost incurred by the increased computation complexity can make up for the cost saved by the decreased computation complexity due to the forbidden repeated search.“. Please refer to the “Computational complexity and limitation” section. (in purple)
Point 12: How about the storage of solutions?
Response 12: (In Tabu-Search-Based Enhanced Rosenbrock Algorithm section) We've rewritten the Tabu Table section. Please refer to the second paragraph in the “Tabu-Search-Based Enhanced Rosenbrock Algorithm” section. (in purple)
Point 13: How does the proposed algorithm deal with larger neighbourhood? What are the restrictions/limitations/time? Can the authors elaborate on it?
Response 13: (In Tabu-Search-Based Enhanced Rosenbrock Algorithm section) We added the following statement “The neighborhood radius of 1 as an example is discussed in this paper. In TS, defined neighborhoods are not all searched, but can be a part of them. We can increase the scope of exploration by increasing the radius of the neighborhood, but the search complexity increases. To this end, some methods are adopted to select some neighborhood explorations, such as random selection, or steepest descending direction selection.“. Please refer to the “Neighborhood definition” section. (in purple)
Point 14: The resolution/quality of the plotted graphics is quite low. Please try to use vector graphic images.
Response 14: (In Experimental and Result Evaluations section) We've already used vector graphic images. Please refer to the “Experimental and Result Evaluations” section. (in purple)
Best wishes
Authors: Xiaoyu Li, Changyin Sun, Fan Jiang
Reviewer 2 Report
The paper presents an improved Rosenbrock algorithm for antenna array beam scanning by using tabu search. The simulated results show some improved performance as shown in FIg. 3 in terms of received signal to noise ratio. It will be better to also compare other properties among different algorithms such as complexity, calculation time, memory usage and so on. The results are only done in the simulation. It will be better to have some actual experimental results in section 6.
Author Response
Response to Reviewer 2 Comments
Point 1: Extensive editing of English language is also strongly recommended to increase the manuscript’s quality. 

Response 1: We have edited the language and style of the article extensively. (in red)
Point 2: It will be better to also compare other properties among different algorithms such as complexity, calculation time, memory usage and so on. The results are only done in the simulation. It will be better to have some actual experimental results in section 6. 

Response 2: (In Tabu-Search-Based Enhanced Rosenbrock Algorithm section) We added “Computational complexity and limitation” section to explain the Point 2.
(In Experimental and Result Evaluations section) We have further added “Complexity Analysis” section to compare the searching complexity of different algorithms. (in purple)
Best wishes
Authors: Xiaoyu Li, Changyin Sun, Fan Jiang
Reviewer 3 Report
The paper presents an interesting optimization problem of finding the best beam pair in millimeter wave communication systems. The paper is carefully prepared. The problem model and the beam training algorithms are precisely described. Tabu search is applied to improve a state-of-the-art beam-searching method combining Rosenbrock algorithm and simulated annealing. The evaluation shows that the proposed method can significantly improve the SNR at the receiver.
Comments:
- Rosenbrock should be written with a capital initial.
- The number of receiver antennas is denoted by M_r in the text while they are denoted by N_t in the related figure.
- Identation could make the description of the algorithms more clear.
Author Response
Response to Reviewer 3 Comments
Point 1: Extensive editing of English language is also strongly recommended to increase the manuscript’s quality. 

Response 1: We have edited the language and style of the article extensively. (in red)
Point 2: Rosenbrock should be written with a capital initial. 

Response 2: We’ve modified ‘rosenbrock’ to 'Rosenbrock'. (in red)
Point 3: The number of receiver antennas is denoted by M_r in the text while they are denoted by N_t in the related figure.
Response 3: We’ve modified ‘N_t’ to ' M_r '. Please refer to Figure 1. (in purple)
Best wishes
Authors: Xiaoyu Li, Changyin Sun, Fan Jiang